# Tetrafluoroisopropylation of alkenes and alkynes enabled by photocatalytic consecutive difluoromethylation with CF$_2$HSO$_2$Na

Yuwei Hong[1,2], Jiayan Qiu[1,2], Zhenzhen Wu[1], Sangxuan Xu[1], Hanliang Zheng ®[1]✉ & Gangguo Zhu ®[1]✉

Direct assembly of complex fluorinated motifs from simple fluorine sources is an attractive frontier of synthetic chemistry. Reported herein is an unconventional protocol for achieving tetrafluoroisopropylation by using commercially available CF$_2$HSO$_2$Na as a convenient source of the tetrafluoroisopropyl [(CF$_2$H)$_2$CH] group, which finds widespread applications in life science and material science. Visible-light-induced hydrotetrafluoroisopropylation of alkenes and carbotetrafluoroisopropylation of alkynes have been thus developed. Various structurally diverse α-tetrafluoroisopropyl carbonyls and cyclopentanones are selectively constructed under mild conditions. A photocatalytic triple difluoromethylation cascade, driven by consecutive reductive radical/polar crossover processes, leads to the direct assembly of a tetrafluoroisopropyl moiety from CF$_2$HSO$_2$Na. This C$_1$-to-C$_3$ fluoroalkylation protocol provides a practical strategy for the rapid construction of polyfluorinated compounds that are otherwise difficult to access, thus significantly enhancing the boundary of fluoroalkylation chemistry.

The incorporation of fluoroalkyl (R$_f$) moieties into organic compounds is a common and useful means to tune the physical, chemical, and biological properties, which are important for discovery of pharmaceuticals, agrochemicals, and advanced materials[1–4]. For example, the difluoromethyl (CF$_2$H) group is a lipophilic H-bond donor and can be used as the biological isostere of a hydroxy, a thiol, and an amide[5,6]. Consequently, the past decades have witnessed a rapid development in the synthesis and application of difluoromethyl-containing molecules[7]. Among these, tetrafluoroisopropylated compounds [(CF$_2$H)$_2$CX (X = H, OH, halide, etc.)] are attracting more and more attention, because of their increased electron-withdrawing capability, hydrogen-bonding interaction, metabolic stability, and hydrophobicity enabled by the *gem*-difluoromethyl substitution[8–12]. Notable examples include the discovery of pesticide, protein stabilizer, ASH1L

inhibitor, and potassium channel opener (Fig. 1a). Additionally, they have been widely used for the construction of functional materials[13]. More importantly, the two terminal C-H bonds in (CF$_2$H)$_2$CX provide more sites for biodegradation, which can avoid the health and environmental concerns about PFAS (per- and polyfluoroalkyl substances)[14]. Despite these intriguing applications, there is a dearth of methods for the efficient synthesis of tetrafluoroisopropyl motifs[15–19]. It was reported that the (CF$_2$H)$_2$CH group could be constructed via a Wittig reaction followed by hydrogenation (Fig. 1b)[15], however, the use of highly volatile 1,1,3,3-tetrafluoroacetone (CF$_2$HCOCF$_2$H) has hindered its application. An alternative procedure relied on the electroreductive double hydrodifluoromethylation of terminal aryl alkynes with Hu reagent, 2-PySO$_2$CF$_2$H, which suffers from low yields and limited substrate scope[16]. Therefore, the development of efficient and general

[1]Key Laboratory of the Ministry of Education for Advanced Catalysis Materials, College of Chemistry and Materials Science, Zhejiang Normal University, 688 Yingbin Road, Jinhua 321004, P. R. China. [2]These authors contributed equally: Yuwei Hong, Jiayan Qiu. ✉e-mail: hanliang@zjnu.edu.cn; gangguo@zjnu.cn

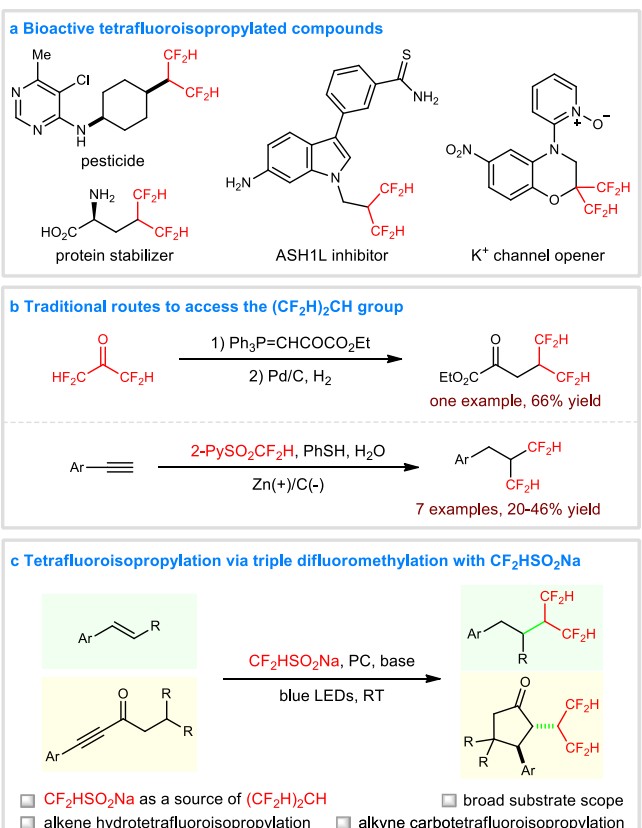

**Fig. 1 | Background and summary of this work. a** Bioactive tetra-fluoroisopropylated compounds. **b** Traditional routes to access the (CF$_2$H)$_2$CH group. **c** Tetrafluoroisopropylation via triple difluoromethylation with CF$_2$HSO$_2$Na.

protocols for accessing tetrafluoroisopropylated compounds is still highly desirable.

Fluoroalkylation is a powerful tool for the expedient synthesis of fluorinated building blocks[20–35]. Compared with the well-investigated trifluoromethylation and difluoromethylation, the tetra-fluoroisopropylation has not been achieved yet, although it represents an ideal method for the direct elaboration of tetrafluoroisopropyl-containing scaffolds. The obstacle comes from the lack of tetra-fluoroisopropylating reagents. As a continuation of our recent interest in radical fluoroalkylations[36–39], we report here an unconventional strategy for tetrafluoroisopropylation, in which commercially available and inexpensive sodium difluoromethanesulfinate (CF$_2$HSO$_2$Na)[40–43] is used as a convenient source of the tetrafluoroisopropyl group, via a visible light photocatalytic[44–47] triple difluoromethylation cascade (Fig. 1c). The robustness of this method is well demonstrated by the successful realization of both hydrotetrafluoroisopropylation of alkenes and carbotetrafluoroisopropylation of alkynes, producing a number of α-tetrafluoroisopropyl carbonyls and *trans*-α-tetra-fluoroisopropyl-β-aryl-cyclopentanones in promising yields with excellent regio- and diastereoselectivity under mild conditions.

## Results

### Reaction optimization
Initial optimizations were conducted with alkene **1a**, CF$_2$HSO$_2$Na (**2a**, 4.0 equiv), and 1,2,3,5-tetrakis(carbazol-9-yl)−4,6-dicyanobenzene (4CzIPN, 5 mol%) in MeCN at 25 °C under irradiation with 24 W blue LEDs for 24 h. Unfortunately, only trace amounts of α- and β-CF$_2$H-substituted esters **4** and **5** were formed (Table 1, entry 1). We conjectured that a suitable base might promote the dehydrofluorina-tion of **4** and thus facilitate the subsequent difluoromethylation.

Consequently, a set of bases were screened with 4CzIPN as the pho-tocatalyst (entries 3–6). Among these, LiOH was the most efficient and α-tetrafluoroisopropyl ester **3** was obtained in 20% yield. Switching the photocatalyst from 4CzIPN to 1,3-dicyano-2,4,5,6-tetrakis(*N,N*-diphe-nylamino)-benzene (4DPAIPN) increased the yield to 25% (entry 7). Other solvents such as DMF and DMSO proved to be less effective (entries 8 and 9). A better yield (52%) was obtained when H$_2$O (10 equiv) was added (entry 10). Increasing the loadings of LiOH and 4DPAIPN further improved the reaction efficiency, providing **3** in 79% yield upon isolation (entries 11 and 12). The stereochemistry of C-C double bonds of **1** has little impact on this hydrotetra-fluoroisopropylation process, as demonstrated by the efficient pro-duction of **3** from *cis*-ethyl cinnamate (entry 13).

### Examination of substrate scope
With the optimized conditions established, the scope of this photo-catalytic hydrotetrafluoroisopropylation reaction was investigated with a panel of functionalized alkenes, and the results are summarized in Fig. 2. Halogen atoms, such as fluorine (**8** and **20**), chlorine (**9** and **21**), and bromine (**10**), were well tolerated under standard conditions, providing opportunities for further functionalizations. Although slightly modified conditions, i.e., replacing LiOH with Cs$_2$CO$_3$, were required in some cases, both strong electron-donating groups like OMe (**12**) and electron-withdrawing groups, such as Ac (**14**), CF$_3$ (**15**), and CN (**16**), remained intact and produced α-tetrafluoroisopropyl esters in satisfactory yields. A methyl group at the *para-* and *ortho*-positions of the aryl ring gave the desired products **18** and **19** in comparable yields (74% and 70%). Notably, a broad range of hetero-aryls, namely 2,3-dihydrobenzofuran (**22**), dibenzo[*b,d*]furan (**23**), pyridine (**24–26**), benzo[*d*]thiazole (**27**), and thiophene (**28**), were all compatible. In addition to disubstituted alkenes, the reaction of tri-substituted olefins proceeded efficiently to furnish **29** and **30** in 65% and 75% yield, respectively. Variation of the R$^3$ group was then con-ducted. Alkenes activated by Ac, P(O)(OEt)$_2$, and CN worked well for this hydrotetrafluoroisopropylation process (**31-33**), while the sub-stitution with NO$_2$ and Bz was unsuccessful (**34** and **35**). In contrast, (*E*)-ethyl 3-cyclohexylacrylate was an ineffective substrate, probably due to the absence of spin delocalization to aryl groups. Given the increasing importance of deuteriodifluoromethylated[48] compounds in pharmaceutical and agrochemical industries, we examined the feasi-bility of assembling a bis(deuteriodifluoromethyl) [(CF$_2$D)$_2$CD] unit from CF$_2$DSO$_2$Na (**2b**). To our delight, the reaction occurred smoothly to afford polydeuterated product **36** in 80% yield. The application of this method for late-stage elaboration of biologically active molecules was conducted as well. Highly functionalized alkenes, derived from ibuprofen (**37**), L-menthol (**38**), estrone (**39**), and empagliflozin (**40**), were successfully transformed to the corresponding α-tetrafluoroisopropyl esters in moderate to high yields.

On the basis of our previous works on 5-endo carbocyclization of alkynes[49–53], we set about to explore the possibility of carbotetra-fluoroisopropylation of alkynyl ketones via a 5-endo-trig ring closure. Screenings (see Supplementary Table 1 in the Supplementary Infor-mation for details) showed that treatment of ynone **41a** (R$^1$ = Ph, R$^2$/R$^3$ = Me) with **2a** (4.0 equiv), Cs$_2$CO$_3$ (2.0 equiv), and 4CzIPN (2 mol %) in MeCN at 25 °C under irradiation with 24 W blue LEDs for 18 h served as the optimal conditions. As a result, *trans*-α,β-disubstituted cyclopentanone **42** was obtained in 86% yield with exclusive diaster-eoselectivity (>20:1 dr, Fig. 3). The structure of **42** was determined by the X-ray crystallography of its hydrazone derivative[54].

This visible-light-induced carbotetrafluoroisopropylation process appeared to be quite general. Various functional groups, such as F (**43**), Cl (**44, 50**, and **51**), Br (**45**), OMe (**46**), CO$_2$Et (**47**), CN (**48**), and OTBS (**58**), were accommodated to form the corresponding cyclopenta-nones in medium to excellent yields with exceptional diastereoselec-tivities. Substrates with pyridine and thiophene substituents worked

**Table 1 | Optimization of reaction conditions**

| Entry[a] | PC | Base | Solvent | Yield of 3/4/5/6/7 (%)[b] |
|---|---|---|---|---|
| 1 | 4CzIPN | none | MeCN | 0/3/2/0/0 |
| 2[c] | 4CzIPN | none | MeCN | 0/3/5/0/0 |
| 3 | 4CzIPN | K₃PO₄ | MeCN | 0/55/14/0/0 |
| 4 | 4CzIPN | K₂CO₃ | MeCN | 0/60/14/0/0 |
| 5 | 4CzIPN | Cs₂CO₃ | MeCN | 4/85/11/0/0 |
| 6 | 4CzIPN | LiOH | MeCN | 20/65/14/0/0 |
| 7 | 4DPAIPN | LiOH | MeCN | 25/66/8/0/0 |
| 8 | 4DPAIPN | LiOH | DMF | 5/30/16/0/0 |
| 9 | 4DPAIPN | LiOH | DMSO | 7/44/15/0/0 |
| 10[d] | 4DPAIPN | LiOH | MeCN | 52/41/6/0/0 |
| 11[d,e] | 4DPAIPN | LiOH | MeCN | 76/13/8/0/0 |
| 12[d,f] | 4DPAIPN | LiOH | MeCN | 86(79)[g]/5/8/0/0 |
| 13[h] | 4DPAIPN | LiOH | MeCN | 82(72)[g]/7/9/0/0 |

[a]Reaction conditions: **1a** (0.2 mmol), **2a** (0.8 mmol), PC (5 mol%), base (0.4 mmol), solvent (2 mL), blue LEDs, 25 °C, 24 h.
[b]The ¹⁹F NMR yield with para-fluoroiodobenzene as an internal standard.
[c]H₂O (2.0 equiv) was added.
[d]H₂O (10 equiv) was added.
[e]Base (0.8 mmol) was used.
[f]PC (10 mol%) was used.
[g]Isolated yield.
[h]Cis-ethyl cinnamate was used instead of **1a**. PC photocatalyst. 4CzIPN 1,2,3,5-tetrakis(carbazol-9-yl)-4,6-dicyanobenzene. 4DPAIPN 1,3-dicyano-2,4,5-tetrakis(N,N-diphenylamino)-benzene. MeCN acetonitrile. DMF N,N-dimethylformamide. DMSO dimethylsulfoxide.

**Fig. 2 | Scope of the photocatalytic hydrotetrafluoroisopropylation of alkenes.** [a]Reaction conditions: **1** (0.2 mmol), **2a** (0.8 mmol), 4DPAIPN (10 mol%), LiOH (0.8 mmol), $H_2O$ (10 equiv), MeCN (2 mL), 25 °C, 24 W blue LEDs, 24 h. [b]$Cs_2CO_3$ (0.8 mmol) and 36 h were used. [c]**2b** was used instead of **2a**. Dr diastereoisomer ratio. RT room temperature.

well in this reaction, producing **54** and **55** in good yields. The generation of spirocyclopentanones was feasible, as demonstrated by the efficient synthesis of **56** and **57**. Gratifyingly, the reaction could be extended to the diastereoselective construction of synthetically challenging bicyclic framework **59**, which bears four contiguous stereocenters. Alkyl ynones (R[1] = alkyl, not shown in Fig. 3) did not participate in this reaction, presumably due to the lack of spin delocalization to aryl groups. Likewise, this process was amenable to construct complex scaffolds stemmed from ibuprofen (**60**), clofibrate (**61**), amino acid (**62**), and estrone (**63**), making it an appealing protocol for the concise synthesis of biologically active compounds.

## Mechanistic investigations

A set of control experiments were then performed to clarify the mechanism of this hydrotetrafluoroisopropylation process (Fig. 4). The model reaction was completely inhibited by the addition of 2,2,6,6-tetramethylpiperidinooxy (TEMPO), which is consistent with a radical pathway. In the presence of excess $D_2O$ (>98% D), [$D_2$]-**3** was formed in 70% yield. Incorporation of 65% and 95% D at the α- and β-carbon atoms, respectively, implies that carbanion formation is possible at these two positions. Under standard conditions, α-difluoromethyl ester **4**, monofluoroalkene **6**[55], and difluoromethylated alkene **7** could be converted to the tetrafluoroisopropylated product **3** in high yields, supporting the involvement of these compounds as key reaction intermediates.

Based on these results, a possible mechanism for the alkene hydrotetrafluoroisopropylation, encompassing three consecutive photocatalytic cycles, is proposed in Fig. 5a, with **1a** and **2a** as

representative substrates. In all the catalytic cycles, single electron transfer (SET) between the excited photocatalyst (PC*) and **2a** produces •$CF_2H$ and a reduced photocatalyst (PC•−). Spin delocalization to the benzene ring enables a regioselective addition of •$CF_2H$ to the α-carbon atom of **1a**. Benzyl radical **A** is thus formed, which can be reduced by PC•− to yield benzyl carbanion **B**, thereby closing the first catalytic cycle. Driven by formation of a more stable carbanion, a formal 1,2-proton transfer (1,2-PT), probably via sequential protonation of the benzyl carbanion and base-promoted deprotonation at the α-position, transforms **B** to α-$CF_2H$-substituted carbanion **C**. A rapid β-fluoride elimination[56,57] then generates monofluoroalkene **6**. In the second photocatalytic cycle, addition of •$CF_2H$ to **6**, and subsequent SET reduction and β-fluoride elimination, produce difluoromethylated alkene **7**. Formation of intermediates **6** and **7** could also be confirmed by HRMS analysis. In the third photoredox cycle, addition of •$CF_2H$ to **7** results in the generation of α-carbonyl radical **F**. A subsequent SET reduction by PC•− and protonation yield **3** as the final product.

On the other hand, initiated by a cascade radical addition to ynones, 1,5-hydrogen atom transfer (1,5-HAT), 5-endo-trig closure, and SET reduction that was developed in our previous works[51], a similar α-$CF_2H$-substituted carbanion **K** is formed (Fig. 5b), which can undergo two more difluoromethylations, as described above, to form the carbotetrafluoroisopropylation product **42**. As such, we have developed a $C_1$-to-$C_3$ tetrafluoroisopropylation with commercially accessible $CF_2HSO_2Na$ as the only fluorine source, successfully avoiding the preparation of complex fluoroalkylating reagents. Of note, pioneered by Hu, Zhang, and others, the controllable fluorocarbon chain

**Fig. 3 | Scope of the photocatalytic carbotetrafluoroisopropylation of alkynyl ketones.** Reaction conditions: **41** (0.2 mmol), **2a** (0.8 mmol), 4CzIPN (2 mol%), $Cs_2CO_3$ (0.4 mmol), MeCN (2 mL), 25 °C, 24 W blue LEDs, 18 h. Unless otherwise stated, the desired products were obtained with >20:1 dr selectivity. TBS *tert*-butyldimethylsilyl.

elongation (CFCE)[58–67] can also assemble advanced fluoroalkyl groups, such as tetrafluoroethyl ($CF_2CF_2$), pentafluoroethyl ($CF_2CF_3$), trifluoroalkenyl ($CF=CF_2$), pentafluorocyclopropyl ($CFCF_2CF_2$), and tetrafluoropropanoyl ($COCF_2CF_2H$) motifs, from $C_1$ synthons, such as the Ruppert–Prakash reagent ($TMSCF_3$)[58–61], $TMSCF_2Br$[62–64], and $BrCF_2PO(OEt)_2$[65]. In contrast to the CFCE methodology proceeding via a difluorocarbene pathway, the radical assembly strategy developed here allows for a mechanistically distinct paradigm for achieving the challenging but significant $C_1$-to-$C_n$ fluoroalkylation.

### Synthetic applications

The synthetic utility was investigated (Fig. 6). The photocatalytic hydrotetrafluoroisopropylation of **1zh** proceeded efficiently to afford α-tetrafluoroisopropyl ester **64** in 71% yield. A subsequent hydrolysis furnished α-tetrafluoroisopropyl acid **65** in 95% yield. Given the significant bioactivity of its parent compound **66**[68], we evaluated the biological activity of compound **65**. It did exhibit a potent peroxisome proliferators-activated receptor α (PPARα) transactivation activity ($EC_{50} = 4.2\,\mu M$), albeit with a lower activity than **66** (see Supplementary Fig. 6 in the Supplementary Information for details).

### Discussion

In summary, visible light photocatalytic tetrafluoroisopropylations, including hydrotetrafluoroisopropylation of alkenes and carbotetrafluoroisopropylation of alkynes, are accomplished by using

$CF_2HSO_2Na$ as a precursor of the tetrafluoroisopropyl group. The reactions allow for facile, efficient, and highly selective construction of α-tetrafluoroisopropyl carbonyls and *trans*-α,β-disubstituted cyclopentanones from readily accessible starting materials. Mechanistic investigations indicate that the key to this $C_1$-to-$C_3$ fluoroalkylation lies in three consecutive reductive radical/polar crossover processes trapped by two β-fluoride eliminations and one protonation. This radical assembly strategy opens up a pathway for the concise synthesis of complex fluorinated molecules that are difficult to obtain via traditional methods. We anticipate that direct assembly of other fluorinated and even non-fluorinated architectures via this strategy will be achieved in the near future.

## Methods

### Procedure for the photocatalytic hydrotetrafluoroisopropylation of alkenes

To a mixture of $CF_2HSO_2Na$ (112 mg, 0.8 mmol), 4DPAIPN (15.9 mg, 0.02 mmol), $H_2O$ (36.0 mg, 2.0 mmol), and LiOH (19.2 mg, 0.8 mmol) in 2 mL of MeCN was added **1a** (35.2 mg, 0.2 mmol) under a nitrogen atmosphere. After 24 h of irradiation at a distance of -2 cm with 24 W of blue LEDs (PINO® lamps, 100% light intensity) at 25 °C, the reaction mixture was quenched with water, extracted with EtOAc, washed with brine, dried over anhydrous $Na_2SO_4$, and concentrated. The resulting residue was purified via column chromatography on silica gel to afford the desired product.

## Procedure for the photocatalytic carbotetrafluoroisopropylation of alkynes

To a mixture of $CF_2HSO_2Na$ (112 mg, 0.8 mmol), 4CzIPN (3.2 mg, 0.004 mmol) and $Cs_2CO_3$ (130 mg, 0.4 mmol) in 2 mL of MeCN was added **41a** (35.2 mg, 0.2 mmol) under a nitrogen atmosphere. After 18 h of irradiation at a distance of ~2 cm with 24 W of blue LEDs (PINO® lamps, 100% light intensity) at 25 °C, the reaction mixture was quenched with water, extracted with EtOAc, washed with brine, dried over anhydrous $Na_2SO_4$, and concentrated. The resulting residue was purified via column chromatography on silica gel to afford the desired product.

## Evaluation of the PPARα transactivation activities

The transactivation activities on PPARα of compounds **65** and **66** were assessed using the Stop & Glo reagent, according to the manufacturer's instructions. HEK293 cells, purchased from American Type

**Fig. 4 | Mechanistic studies.** TEMPO 2,2,6,6-tetramethylpiperidinooxy.

**Fig. 6 | Synthetic applications.** Reaction conditions: **a** see Table 1, entry 11. **b** (i) NaOH, EtOH/H₂O, 50 °C; (ii) 2.0 M HCl. EC₅₀ median effective dose.

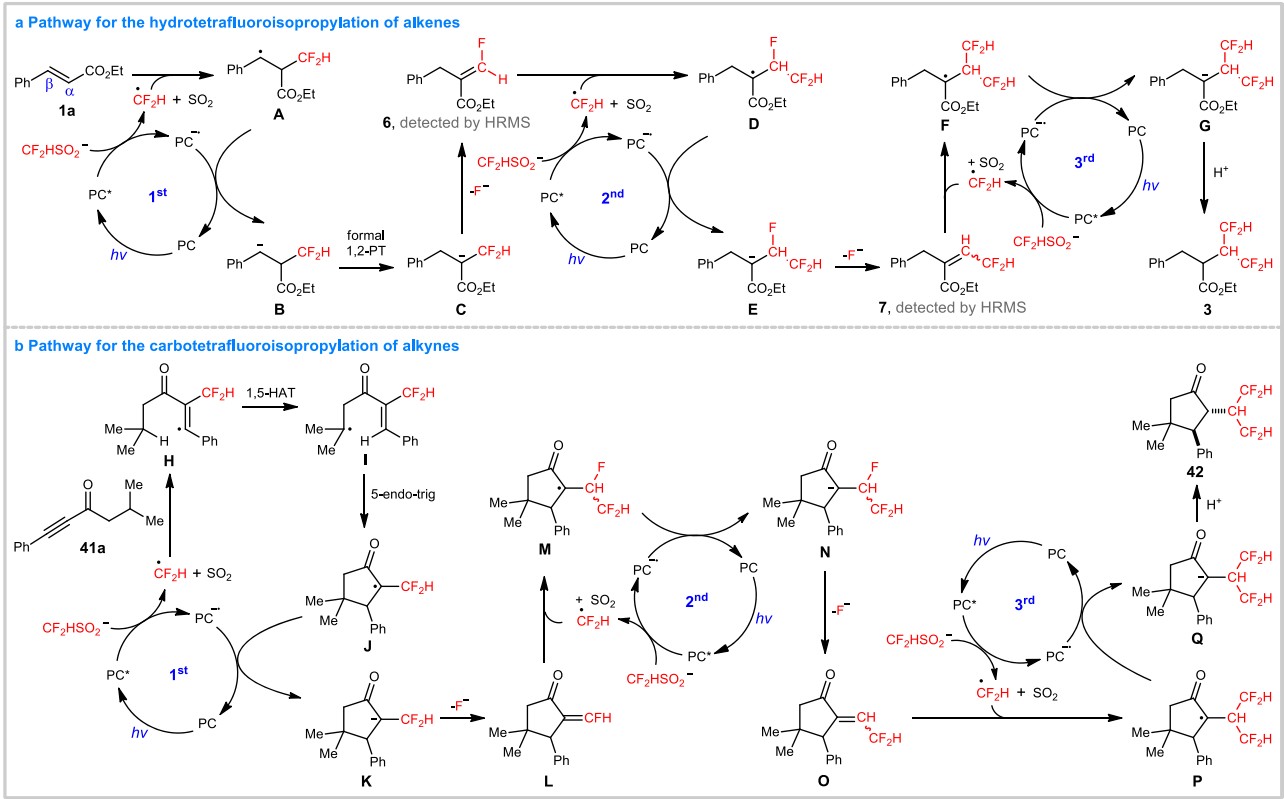

**Fig. 5 | Proposed mechanism. a** Pathway for the hydrotetrafluoroisopropylation of alkenes. **b** Pathway for the carbotetrafluoroisopropylation of alkynes. HRMS high resolution mass spectrometry. 1st the first photocatalytic cycle. 2nd the second photocatalytic cycle. 3rd the third photocatalytic cycle.

Culture Collection (ATCC) with a catalog number of CRL-1573, were authenticated by Short Tandem Repeat test, then seeded into 96-well plates at a density of $8 \times 10^4$ cells/well in 90 μL of cell seeding medium (97% DMEM without phenol red, 2% charcoal stripped FBS and 1% GlutaMax) together with 10 μL transfection reagent (PPARα 1.079 mg/mL and pGL4.35 1.317 mg/mL). Compounds **65** and **66** were prepared 4-fold serial dilution with DMSO starting at 400 μM, 8 points in total, then transferred 500 nL to the compound plate using an Echo liquid handler. 10-Fold dilutions of the compounds with 40 μL culture medium (88% DMEM with phenol red, 10% FBS, 1% P/S and 1% GlutaMax) followed by transferring 10 μL to cell plates, which were placed in an incubator at 37 °C for 24 h. After removing 50 μL medium from each well, 50 μL luciferase assay reagent was added to the assay plate, followed by shaking at 25 °C for 20 min. The data were read on an Envision (Perkin Elemer: Envision 2105), then analyzed using XL-fit software (Supplier: ID Business Solutions Ltd., Software version: XL fit 5.0). Effect% = (Sample value−LC)/(HC−LC) × 100.

### Reporting summary

Further information on research design is available in the Nature Portfolio Reporting Summary linked to this article.

## Data availability

The data supporting the results of this study, including optimization studies, experimental procedures, characterization of new compounds, and mechanistic studies, are provided within the paper and its Supplementary Information. The X-ray crystallographic coordinates for structures reported in this study have been deposited at the Cambridge Crystallographic Data Centre, under deposition numbers CCDC 2355597 (**64**) and 2323408 (**67**). Copies of the data can be obtained free of charge via https://www.ccdc.cam.ac.uk/structures/. All data are available from the corresponding author upon request.

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

## Acknowledgements

We thank the Natural Science Foundation of Zhejiang Province (LZ20B020001 to G.Z., and LY23B020004 to H.Z.), the Ten Thousand Talents Plan of Zhejiang Province (2020R52021 to G.Z.), the Leading Innovative and Entrepreneur Team Introduction Program of Zhejiang Province (2022R01007 to G.Z., and others), and the National Natural Science Foundation of China (22371262 to G.Z., 22071218 to G.Z., and 22203076 to H.Z.) for financial support.

## Author contributions

G.Z. conceived the idea. Y.H., J.Q., Z.W., and S.X. conducted the experiments. H.Z. and G.Z. co-wrote the paper. All the authors discussed the results and commented on the manuscript.

## Competing interests

The authors declare no competing interests.
