## [Peer Review File · Nature Communications]

Tetrafluoroisopropylation of alkenes and alkynes enabled by photocatalytic consecutive difluoromethylation with CF₂HSO₂NaReviewers' Comments:

Reviewer #1:

Remarks to the Author:

In this manuscript, Zhu, Zheng, and coworkers described the development of a visible-light-induced hydrotetrafluoroisopropylation of alkenes and carbonyl tetrafluoroisopropylation of alkynes, with the use of commercially available $\text{CF}_2\text{HSO}_2\text{Na}$ as the radical precursor. The reaction exhibits good substrate scope and functional group tolerance, preparing structurally diverse α -tetrafluoroisopropyl carbonyl and cyclopentanone motifs with ease. In addition, preliminary experimental investigations have been performed to clarify the possible reaction mechanism. The manuscript as well as the supporting materials were well prepared. However, several factors diminish the impact and innovation of this study to warrant publication in high quality journals such as *Nat. Commun.*

First, photocatalyzed tri/difluoromethylation reactions of electron-deficient internal alkenes or alkynes have been extensively reported in the last decades, with the utilization of diverse tri/difluoromethyl radical precursors. Despite the unprecedented success for the three consecutive photocatalytic cycles, this reaction did not show sufficient novelty in terms of reaction mechanism. In my opinion, the realization of this reaction is more like the side process observed by the authors when exploring the reaction of olefins with difluoromethyl radical source, rather than a well-designed reaction process, and the significance of this reaction process is limited.

Second, I agree this protocol is a nice piece of synthetic chemistry, but it suffers from the authors trying too hard to find reasons that promote the utility of the afforded $[(\text{CF}_2\text{H})_2\text{CH}]$ group-containing products. I have carefully searched the molecules containing tetrafluoroisopropyl unit provided by the authors in Figure 1a, but none of those show excellent biological activity or have the potential to be a molecular drug. Moreover, using the developed synthetic method cannot synthesize these molecules, only tetrafluoroisopropyl molecules with an α -carbonyl group. If the authors want to demonstrate that $[(\text{CF}_2\text{H})_2\text{CH}]$ unit can replace isopropyl with better biological activity, they should carry out some more in-depth biochemical research, for example, if the biological test shows that compound 61 showed greater advantages than 62 (Fig. 6). In my opinion the current results are not very convincing.

Overall, this work can be regarded as improvement of current CF_2H radical-mediated research, but lacks the conceptual novelty and significance. For these reasons, I don't recommend its publication in *Nat. Commun.* Against the background of the vast previous literature, the present article would be better suited in a more specialized journal.

Other concerns with the manuscript are as follows:

In the mechanistic pathway shown in Fig. 5a, the 1,2-proton transfer (1,2-PT) step should be reconsidered, otherwise the authors would not afford product $[\text{D}_2]$ -3 with 95% D incorporation in the presence of 20 equiv. of D_2O (see Fig. 4).

Reviewer #2:

Remarks to the Author:

The manuscript by Zhu and coworkers describes a novel tetrafluoroisopropylation reaction using the difluoromethylation reagent $\text{NaSO}_2\text{CF}_2\text{H}$. This method enables the installation of a $(\text{CF}_2\text{H})_2\text{CH}$ group on $\text{C}(\text{sp}^3)$ positions in various substrates, such as styrenes and alkynyl ketones. Given the unique tetrafluoroisopropylation strategy and relatively broad scope of this transformation, I believe this work is suitable for publication at *Nat. Commun.* However, I would suggest the authors consider the following suggestions:

1. The majority of styrenes shown in Figure 2 possess an ester functional group. It would be beneficial to explore if other electron-withdrawing groups, such as cyano (CN), are compatible with the reaction conditions. In addition, it seems that all the styrenes in Fig. 2 are trans-alkenes. Can cis-alkenes also work in this transformation?

2. Although a crystal structure is provided for a hydrazone derivative in Fig. 3, obtaining crystal structures for at least one representative product from Fig. 2 is crucial. This is necessary to confirm the structure of compounds reported in Fig. 2

3. The mechanistic studies appear to be weak. The authors have proposed a possible catalytic cycle that involves the addition of CF₂H radicals to alkenes three times. The authors have also shown that compound 4 can be converted to the final product under standard conditions, suggesting that 4 is a possible intermediate. Nonetheless, the authors should also evaluate whether compounds D and G, which are proposed as key intermediates in the catalytic cycle, could also be converted to the final product under reaction conditions.

4. The synthetic utility of this method has been demonstrated by the compound 61, an analog of a bioactive molecule 62. The authors have shown that compound 62 has an EC₅₀ value of 0.29 μM. However, it is not clear what the target protein of compound 62 is. Furthermore, to further highlight the utility of the (CF₂H)₂CH groups in medicinal chemistry, the authors should evaluate the biological activity of compound 61. Admittedly, 61 might or might not show activity toward the target protein, but it will significantly enhance the quality of this paper if compound 61 is bio-active (if not, just include the results in SI).

Reviewer #3:

Remarks to the Author:

The site selective introduction of fluorine functionalities has received increasing attention due to the important applications of organofluorine compounds in pharmaceuticals, agrochemicals, and materials science. Compared to the previous introduction of the CF₃ and CF₂H group, this manuscript describes an unconventional method for tetrafluoroisopropylation using commercially available CF₂HSO₂Na under visible light irradiation. The reaction relies on three consecutive reductive radical/polar crossover processes driven by two β-F eliminations and one protonation. The reaction could provide various hydrotetrafluoroisopropylated products using β-aryl substituted α,β-unsaturated esters with high efficiency. High functional group tolerance was observed in this process, even towards complex molecules. The reaction could also extend to ynones via a 5-endo cyclization, providing a series of α-tetrafluoroisopropylated cyclic ketones with high yields, providing efficient access to these uncommon fluorinated compounds. The synthesis of an analog of bioactive molecules has also demonstrated the impact of the protocol. Overall, this is a very nice strategy to access tetrafluoroisopropylated compounds that are not easy to access through conventional methods. The manuscript can be published in Nat. Commun. after minor revisions.

1) Draw the structure photocatalyst in Table 1.

2) How about aliphatic alkenes? What will happen if the ester group is replaced with other electron-withdrawing groups, such as CN, NO₂, and ArCO?

3) For the difluoromethylation and difluorocarbene elongation reactions, cite Nat. Chem. 2017, 9, 918 and Nat. Chem. 2023, 15, 1064.

To Reviewer 1

Comments: 1. First, photocatalyzed tri/difluoromethylation reactions of electron-deficient internal alkenes or alkynes have been extensively reported in the last decades, with the utilization of diverse tri/difluoromethyl radical precursors. Despite the unprecedented success for the three consecutive photocatalytic cycles, this reaction did not show sufficient novelty in terms of reaction mechanism. In my opinion, the realization of this reaction is more like the side process observed by the authors when exploring the reaction of olefins with difluoromethyl radical source, rather than a well-designed reaction process, and the significance of this reaction process is limited.

Responses: Thanks for the comments! As we known, many landmark achievements in science have come from unexpected results, and a well-designed reaction does not guarantee its novelty and significance. Despite the prevalence of tri/difluoromethylation of alkenes and alkynes, the more challenging tetrafluoroisopropylation processes, are still unknown. The work presented here constitutes the first hydrotetrafluoroisopropylation of alkenes and carbotetrafluoroisopropylation of alkynes. More importantly, it features an unprecedented radical-mediated C1-to-C3 fluoroalkylation, which can avoid the synthesis of complex and difficult-to-access tetrafluoroisopropylating reagents. As such, our method represents a conceptually novel strategy for the realization of unconventional fluoroalkylations, and should be suitable for publication in this journal.

Comments: 2. Second, I agree this protocol is a nice piece of synthetic chemistry, but it suffers from the authors trying too hard to find reasons that promote the utility of the afforded [(CF₂H)₂CH] group-containing products. I have carefully searched the molecules containing tetrafluoroisopropyl unit provided by the authors in Figure 1a, but none of those show excellent biological activity or have the potential to be a molecular drug. Moreover, using the developed synthetic method cannot synthesize these molecules, only tetrafluoroisopropyl molecules with an α -carbonyl group. If the authors want to demonstrate that [(CF₂H)₂CH] unit can replace isopropyl with better biological activity, they should carry out some more in-depth biochemical research, for example, if the biological test shows that compound 61 showed greater advantages than 62 (Fig. 6). In my opinion the current results are not very convincing. Overall, this work can be regarded as improvement of current CF₂H radical-mediated research, but lacks the conceptual novelty and significance. For these reasons, I don't recommend its publication in Nat. Commun. Against the background of the vast previous literature, the present article would be better suited in a more specialized journal.

Responses: As shown in Ref 11, the tetrafluoroisopropylated compound (Fig. 1a, right) was reported to be a **very potent antihypertensive agent** (IC₅₀ = **0.60±0.09 nM**, Guinea-pig portal vein). Therefore, the judgment "none of those show excellent biological activity or have the potential to be a molecular drug" is not solid. The synthetic utility of this method has already been demonstrated by the production of compound **65**. As we known, the carbonyl motif can be easily transformed to other functional groups, and even be removed under suitable conditions, thus rendering this protocol a widely applicable method for the construction of structurally diverse tetrafluoroisopropylated compounds. Of course, we agree with the reviewer that biological investigations might improve the value of this work, as such, the biochemical research of compounds **65** and **66** was conducted. Compound **65** did exhibit a potent human PPAR α transactivation activity (EC₅₀ = 4.2 μ M), albeit with a lower activity than **66** (EC₅₀ = 0.43 μ M). To

summarize, given the realization of an unprecedented radical-mediated C1-to-C3 fluoroalkylation, this work should be suitable for publication in this journal.

Comments: 3. Other concerns with the manuscript are as follows: In the mechanistic pathway shown in Fig. 5a, the 1,2-proton transfer (1,2-PT) step should be reconsidered, otherwise the authors would not afford product [D₂]-**3** with 95% D incorporation in the presence of 20 equiv. of D₂O (see Fig. 4).

Responses: We do agree with the reviewer that [D₂]-**3** could not be obtained with 95% D at the benzyl position if a direct 1,2-proton transfer was involved. As such, “a formal 1,2-proton transfer (1,2-PT), probably via sequential protonation of the benzyl carbanion and base-promoted deprotonation at the α -position” was already used in our previous submission for simplifying the subsequent discussions. It seemed that the reviewer skipped the word “formal” and above explanations.

To Reviewer 2

Comments: 1. The majority of styrenes shown in Figure 2 possess an ester functional group. It would be beneficial to explore if other electron-withdrawing groups, such as cyano (CN), are compatible with the reaction conditions. In addition, it seems that all the styrenes in Fig. 2 are trans-alkenes. Can cis-alkenes also work in this transformation?

Responses: Following the good suggestions, more investigations were conducted. Replacing the ester group with Ac, P(O)(OEt)₂, and CN proved to be successful (Fig. 2, **31-33**). The reactivity of *cis*-ethyl cinnamate was also examined, leading to the formation of **3** in 72% yield (Table 1, entry 13). It indicated that the stereochemistry of styrenes has little impact on this process. The results and related discussions were added accordingly.

Comments: 2. Although a crystal structure is provided for a hydrazone derivative in Fig. 3, obtaining crystal structures for at least one representative product from Fig. 2 is crucial. This is necessary to confirm the structure of compounds reported in Fig. 2.

Responses: Following the good suggestions, the X-Ray data of **64** were added in the Supplementary Information (SI), which undoubtedly confirmed the structure of hydrotetrafluoroisopropylation products.

Comments: 3. The mechanistic studies appear to be weak. The authors have proposed a possible catalytic cycle that involves the addition of CF₂H radicals to alkenes three times. The authors have also shown that compound **4** can be converted to the final product under standard conditions, suggesting that **4** is a possible intermediate. Nonetheless, the authors should also evaluate whether compounds **D** and **G**, which are proposed as key intermediates in the catalytic cycle, could also be converted to the final product under reaction conditions.

Responses: Following the good suggestions, more control experiments were conducted. The reactions of **D** and **G**, currently labeled as **6** and **7**, occurred smoothly to deliver the desired product **3** in high yields (Fig. 4), further confirming the involvement of these two intermediates. The results and discussions were added accordingly.

Comments: 4. The synthetic utility of this method has been demonstrated by the compound **61**, an

analog of a bioactive molecule 62. The authors have shown that compound 62 has an EC50 value of 0.29 μM . However, it is not clear what the target protein of compound 62 is. Furthermore, to further highlight the utility of the $(\text{CF}_2\text{H})_2\text{CH}$ groups in medicinal chemistry, the authors should evaluate the biological activity of compound 61. Admittedly, 61 might or might not show activity toward the target protein, but it will significantly enhance the quality of this paper if compound 61 is bio-active (if not, just include the results in SI).

Responses: Following the good suggestions, the biochemical research of compounds **65** and **66** was conducted. Compound **65** did exhibit a potent human PPAR α transactivation activity ($\text{EC}_{50} = 4.2 \mu\text{M}$), albeit with a lower activity than **66** ($\text{EC}_{50} = 0.43 \mu\text{M}$). The results and related discussions were added in the manuscript as well as SI.

To Reviewer 3

Comments: 1) Draw the structure photocatalyst in Table 1.

Responses: Added as suggested.

Comments: 2) How about aliphatic alkenes? What will happen if the ester group is replaced with other electron-withdrawing groups, such as CN, NO₂, and ArCO?

Responses: Following the good suggestions, we examined the reaction of mentioned substrates. Aliphatic alkenes like (*E*)-ethyl 3-cyclohexylacrylate failed to provide the desired tetrafluoroisopropylated product, presumably due to the absence of spin delocalization to the Ar group. Replacing the ester group with Ac, P(O)(OEt)₂, and CN proved to be successful (Fig. 2, **31-33**), while the substitution with NO₂ and Bz was unfruitful (Fig. 2, **34** and **35**). The results and related discussions were added accordingly.

Comments: 3) For the difluoromethylation and difluorocarbene elongation reactions, cite Nat. Chem. 2017, 9, 918 and Nat. Chem. 2023, 15, 1064.

Responses: Cited as suggested.

Reviewers' Comments:

Reviewer #1:

Remarks to the Author:

The authors have carefully revised the manuscript according to the reviewers' comments. Currently I have no problem with the substrate scope and the reaction mechanism. On the novelty of the work, although I still have a little doubt about the utility of the afforded [(CF₂H)₂CH] group-containing products at present, the authors indeed clearly described and discussed the advantages of this protocol that could enable the facile synthesis of bioactive tetrafluoroisopropylated molecules in comparison to current methods. And I totally agree that many great achievements in science come from unexpected results rather than well-designed reaction. Based on my own consideration as well as other reviewers' opinions, I am prone to believe the work is suitable for the readers of this journal.

Reviewer #2:

Remarks to the Author:

The authors have carefully revised this manuscript according to my suggestions. In the revised manuscript, the authors have shown that compound 65 exhibits moderate transactivation activity. Although the biological activity is lower than the parent molecule, these biological studies demonstrate the potential utilities of these fluorinated molecules. Moreover, additional mechanistic studies further support the proposed mechanism. Although the discovery of this reaction is not through rational design (as everyone can tell), many interesting reactions were discovered by serendipity and the significance of this work should not be underestimated. Therefore, I would like to recommend the publication of this work at Nature Communications.

However, there is a minor issue that the authors should address: When running the checkCif, there is a level B alert in one of the crystal structures.

Reviewer #3:

Remarks to the Author:

The authors have fulfilled the requests raised by the referees. Compared to previous reports, the current approach features synthetic simplicity and efficiency. Given the novelty of the construction of the tetrafluoroisopropyl group, the manuscript is recommended for publication in Nat. Commun. without further revision.

To Reviewer 1

Comments: The authors have carefully revised the manuscript according to the reviewers' comments. Currently I have no problem with the substrate scope and the reaction mechanism. On the novelty of the work, although I still have a little doubt about the utility of the afforded [(CF₂H)₂CH] group-containing products at present, the authors indeed clearly described and discussed the advantages of this protocol that could enable the facile synthesis of bioactive tetrafluoroisopropylated molecules in comparison to current methods. And I totally agree that many great achievements in science come from unexpected results rather than well-designed reaction. Based on my own consideration as well as other reviewers' opinions, I am prone to believe the work is suitable for the readers of this journal.

Responses: Thanks for the comments!

To Reviewer 2

Comments: The authors have carefully revised this manuscript according to my suggestions. In the revised manuscript, the authors have shown that compound 65 exhibits moderate transactivation activity. Although the biological activity is lower than the parent molecule, these biological studies demonstrate the potential utilities of these fluorinated molecules. Moreover, additional mechanistic studies further support the proposed mechanism. Although the discovery of this reaction is not through rational design (as everyone can tell), many interesting reactions were discovered by serendipity and the significance of this work should not be underestimated. Therefore, I would like to recommend the publication of this work at Nature Communications. However, there is a minor issue that the authors should address: When running the checkCif, there is a level B alert in one of the crystal structures.

Responses: Thanks for the careful reviewing! The files for crystal data of the mentioned structure were updated and the B-level alert was removed accordingly.

To Reviewer 3

Comments: The authors have fulfilled the requests raised by the referees. Compared to previous reports, the current approach features synthetic simplicity and efficiency. Given the novelty of the construction of the tetrafluoroisopropyl group, the manuscript is recommended for publication in Nat. Commun. without further revision.

Responses: Thanks for the comments!